# Explaining variations in test ordering in primary care: protocol for a realist review

Claire Duddy, Geoffrey Wong

## ABSTRACT

**Introduction** Studies have demonstrated the existence of significant variation in test-ordering patterns in both primary and secondary care, for a wide variety of tests and across many health systems. Inconsistent practice could be explained by differing degrees of underuse and overuse of tests for diagnosis or monitoring. Underuse of appropriate tests may result in delayed or missed diagnoses; overuse may be an early step that can trigger a cascade of unnecessary intervention, as well as being a source of harm in itself.

**Methods and analysis** This realist review will seek to improve our understanding of how and why variation in laboratory test ordering comes about. A realist review is a theory-driven systematic review informed by a realist philosophy of science, seeking to produce useful theory that explains observed outcomes, in terms of relationships between important contexts and generative mechanisms. An initial explanatory theory will be developed in consultation with a stakeholder group and this 'programme theory' will be tested and refined against available secondary evidence, gathered via an iterative and purposive search process. This data will be analysed and synthesised according to realist principles, to produce a refined 'programme theory', explaining the contexts in which primary care doctors fail to order 'necessary' tests and/or order 'unnecessary' tests, and the mechanisms underlying these decisions.

**Ethics and dissemination** Ethical approval is not required for this review. A complete and transparent report will be produced in line with the RAMESES standards. The theory developed will be used to inform recommendations for the development of interventions designed to minimise 'inappropriate' testing. Our dissemination strategy will be informed by our stakeholders. A variety of outputs will be tailored to ensure relevance to policy-makers, primary care and pathology practitioners, and patients.

**Prospero registration number** CRD42018091986

### Strengths and limitations of this study

► First realist review exploring how, why and in what circumstances variations in test ordering in primary care come about.
► Realist approach embraces complexity, seeking to develop understanding of multiple causes of variation and to explore the role of different contexts.
► Involvement of stakeholders in refining programme theory and disseminating outputs will ensure relevance and applicability.
► Availability and richness of available evidence may limit theory building.

of 'inappropriate' test use usually assess observed test use against chosen guideline standards.[16 17] This approach has limitations, as assessments can only be made wherever guidelines or protocols exist, and will only be as reliable as the guidelines themselves.

This review is concerned with the use of laboratory tests in primary care settings. Our initial focus will be on the National Health Service (NHS) in the UK, but we will endeavour to develop recommendations relevant in other settings and countries, where it is likely that the same mechanisms and contexts produce similar outcomes. The use of such tests in UK primary care is extensive and growing,[15] and is known to vary substantially by region.[13 15] In 2006, the Carter Review reported that 35%–45% of requests for laboratory tests in the UK came from primary care.[18] Although an individual laboratory test may be inexpensive, high volumes mean that overall expenditure is high. The same review estimated that pathology services cost the NHS around £2.5 billion per year.[18]

### Undertesting and overtesting

Although variations in test-ordering practice clearly occur, categorising this practice as undertesting or overtesting can be more difficult. As noted above, existing studies usually rely on assessing test-ordering behaviour against existing guideline or protocol standards. For individual patients, it may only be possible to decide that a particular testing

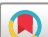

Nuffield Department of Primary Care Health Sciences, Radcliffe Observatory Quarter, Oxford, UK

**Correspondence to**
Claire Duddy;
claire.duddy@phc.ox.ac.uk

## BACKGROUND
### Variation in test ordering

A large number of studies and reports have demonstrated the existence of significant variation in primary and secondary care test-ordering patterns, across many different health systems.[1–15] This variation in practice could be explained by differing degrees of underuse and overuse of diagnostic testing in these different settings. Primary studies and reviews that attempt to assess the extent

decision was 'inappropriate' later, in light of the results and subsequent decisions, and in many cases, this may be impossible to ascertain even then.[19 20] The picture is further complicated by the possibility that undertesting and overtesting may occur simultaneously.[21]

It is clear however that both undertesting and overtesting can have negative consequences for patients. Underutilisation of appropriate tests can result in delayed, missed or incorrect diagnoses and subsequent treatment, and failure to appropriately monitor patients with existing conditions can also result in harm. Uneven access to tests and treatment for different population groups is also a concern.[22–24]

Overtesting is also a problem. Overdiagnosis and consequent overtreatment are increasingly seen as an important source of harm within many healthcare systems. The phenomenon of 'too much medicine' is considered by many to result in direct and indirect harm to individual patients in the form of unnecessary labelling and treatment[25–28] as well as posing a threat to sustainability and equity in healthcare systems, increasing costs[29 30] and diverting resources from the genuinely ill to the 'worried well'.[31]

The increasing interest in this area is reflected in campaigning, including the *BMJ*'s 'Too Much Medicine'[32] (launched in 2002) and 'Choosing Wisely'[33] (launched in the UK in 2016), in a growing number of popular books[34–37] and articles in the mainstream media,[38–41] and in a rapidly growing literature (see online supplementary file). A recent wide-ranging (though not systematic) review[42] drew attention to the large number of 'drivers' of medical overuse that have been identified, but also highlighted the limitations of the existing literature, which is dominated by 'analyses or commentaries'.[42]

Medical overuse, including overtesting, is often considered under the 'overdiagnosis' banner. Precise definitions are contested,[19 43 44] but terminology like 'overdiagnosis' is frequently used broadly by both researchers and activists to cover a wide range of issues. A broad conceptualisation encompasses concerns ranging from the overdetection of harmless cancers during screening (and their subsequent overtreatment)[45] to widening definitions of disease and predisease,[28 46] and many more. The common thread is the identification of medical care that is provided despite 'a low probability of benefiting the person diagnosed'[47] and indeed, the possibility that such care may instead be a source of harm.

'Overtesting' may therefore be defined in these terms, as the use of tests where there is a low probability that test results will benefit the patient. This could be the case where there is a lack of evidence to support the use of a test, the use of tests where their results are unlikely to change subsequent management or unnecessary repeat test ordering. Conversely, 'undertesting' may occur in the opposite circumstances.

Overdiagnosis and overtreatment phenomena are usually quantified only at population level.[44 48] However, outcomes of undertesting and overtesting are the cumulative effect of many individual decisions taken in a variety of circumstances, within the social system of healthcare. A preliminary map of the decisions faced by both patients and doctors in a primary care setting, alongside some important contextual considerations, is provided below in figure 1.

The decision to order tests is an important feature of this process and an over-reliance on testing has been identified as an important early step that may result in a cascade of further testing and intervention, including the potential for overdiagnosis and overtreatment.[35 49 50] In addition, overtesting and its consequences can directly increase anxiety and worry for patients[51–53] and commentators have highlighted the limited capacity of even 'gold-standard' tests in providing definitive diagnostic answers.[44]

## Existing reviews

Two existing systematic reviews assess 'inappropriate' undertesting and overtesting in secondary[14] and primary[12] care settings: both identified significant variation in practice across a wide range of tests and settings. One health technology assessment considers the extent and consequences of routine preoperative testing.[54] In addition, several systematic reviews assess the efficacy of various interventions designed to reduce variability and improve 'appropriateness' of test ordering in a wide variety of settings.[55–67] One review considers a wide range of variables associated with 'test-ordering tendencies'.[68]

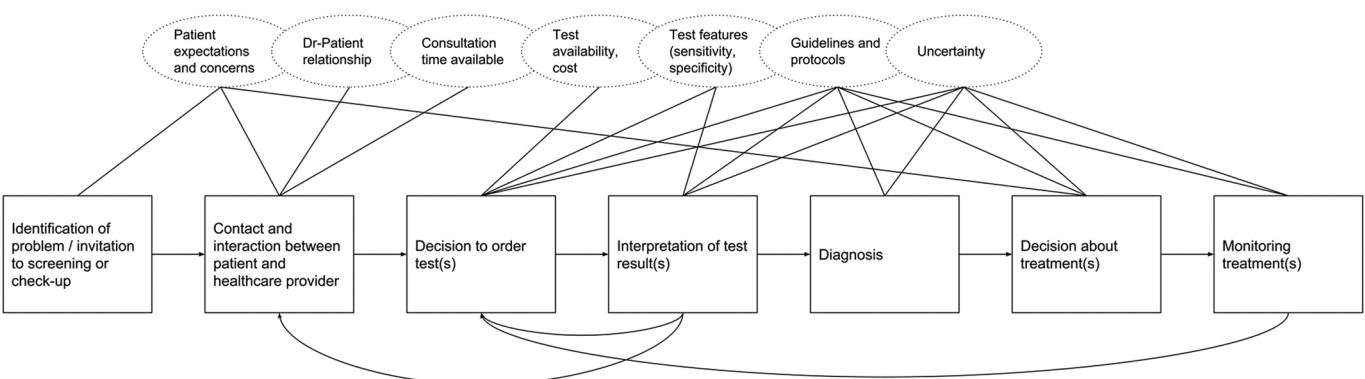

**Figure 1** Steps taken in test-ordering decisions.

No realist reviews on this subject have been found. The wide variation in test-ordering behaviour, and in the outcomes of studies aiming to reduce 'inappropriate' testing indicates that an enquiry into the role of context could have explanatory value for this phenomenon. Patterns of test-ordering behaviour may vary in response to important contextual factors, such as those highlighted in figure 1 above. A number of existing studies have highlighted the wide variety of potential drivers of variation in practice, including clinician and patient characteristics[68 69] and health system characteristics.[2 6 11 68 70] A realist review of the literature will allow consideration of multiple causal mechanisms, sensitivity to context and opening the 'black box'[71] of decision-making in relation to ordering tests.

## Realist review

A realist review (otherwise known as 'realist synthesis') is an interpretive, theory-driven systematic literature review, underpinned by a realist philosophy of science. This philosophy holds that patterns of observed (empirical) outcomes are produced by underlying 'generative' (real) mechanisms, which may or may not be at work in particular contexts.[72] 'Mechanisms' are understood as the causal forces of patterns of observed outcomes (or 'demi-regularities') that have their roots in individual tendencies and reasoning.[73] Causation is 'generative', that is, outcomes in social systems are not the direct result of interventions or simple responses to stimuli, but rather reflect the invisible reasoning and behaviour of actors within those systems.[74] Such reasoning may change (or not) in different contexts, where different resources are available to different actors with different capacities to respond to their circumstances. The realist approach can allow us to go beyond an assessment of those variables associated with a particular outcome, to shed light on the real generative mechanisms that are the underlying causes of observed test use and to highlight the context(s) or conditions in which these mechanisms operate.[75] Contexts and mechanisms are seen as working together to produce outcomes (often expressed as, $C+M \rightarrow O$).[76]

A realist approach may be adopted when there is a need to account for inconsistent outcomes and differences in context, to understand underlying causation and to answer questions that begin 'how', 'why', 'in what circumstances', 'for whom' and so on.[77] Originally proposed as a means to explore the inner workings of similar 'families' of complex social interventions,[73] its utility in helping to 'diagnose' and understand the underlying nature of complex problems has also been established.[78 79] For a glossary of realist terminology, see online supplementary file.

Here, the overall problem of medical overuse and the specific issues of overtesting and undertesting are characterised as 'complex': the literature suggests multiple potential causes operating at different levels, as well as potential emergent effects, whereby (eg) more testing generates even more testing,[25 80] and variable outcomes

exist (eg, undertesting and overtesting coexist in the same healthcare system).[12 81] Decisions to order tests in primary care are made within the context of the interaction between provider and patient; as such there are multiple opportunities for the reasoning and behaviour of both parties to influence the outcome.[82]

Realist inquiry begins (and ends) with a 'programme theory', describing a hypothesis about how an intervention works or how a phenomenon comes about. Realist programme theories are models that describe relationships between important contexts, mechanisms and outcomes, usually presented and described as sequences of 'context–mechanism–outcome configurations' ('CMOCs'). Such configurations aim to explain in which context(s), which mechanism(s) are 'triggered' to produce which outcomes(s). As such, the realist approach is especially useful where outcomes appear to vary with circumstances, seeking to provide explanatory evidence for such variation and offers a means of adjudicating between competing theories and/or refining and improving an initial theory to accommodate multiple explanatory mechanisms.[75]

A realist programme theory should be in the 'middle range', that is, it should be specific enough to permit empirical testing (in this case, against secondary evidence located during the review process), but abstract enough to provide useful, explanatory transferability to other situations where the same mechanisms may be operating.[83]

## REVIEW OBJECTIVES AND DESIGN
### Review objectives

1. Develop a realist programme theory offering explanation(s) for the variation in test ordering in primary care, underpinned by secondary evidence.
2. Make recommendations based on this explanation, to inform the design of existing and new interventions that could help to reduce this problem.

### Review questions

1. How are 'undertesting' and 'overtesting' conceptualised in the literature?
2. In what contexts do primary care doctors order 'unnecessary' tests?
3. In what contexts do primary care doctors fail to order 'necessary' tests?
4. What mechanisms are at work in these different contexts that underlie test-ordering behaviour and generate these outcomes?

The review will be conducted according to Pawson's five stages[84 85] which outline the processes by which an initial programme theory will be developed, evidence gathered and refinements to the theory made. The Realist and Meta-narrative Evidence Syntheses: Evolving Standards (RAMESES) quality[86] and reporting[87] standards will be followed. Figure 2 summarises the overall project design, and more details on each step are provided below.

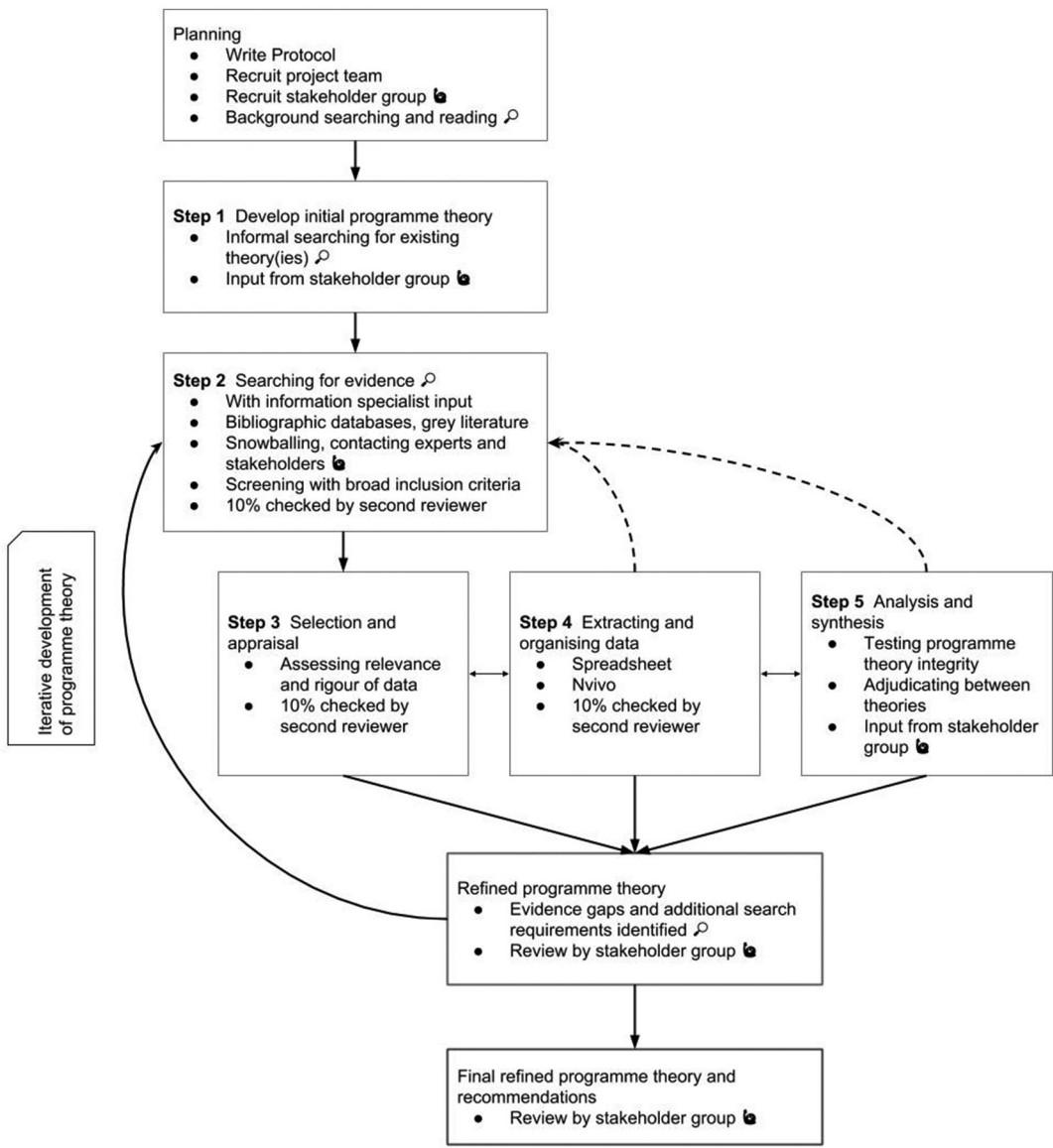

**Figure 2** Review project design.

A 'guiding principle' of the realist approach is the maintenance of transparency of methods and decision-making throughout the review.[87] Such transparency ensures that the iterative nature of the research is made clear and that decisions taken in consultation with stakeholders and within the project team are fully explained and justified. Such decisions determine the direction and focus of the project, as well as guiding the extent and direction of literature searching, and the analysis and synthesis themselves.

### Stakeholder involvement

Following an established approach,[79] a diverse stakeholder group will be recruited at the beginning of the project. This group will include, for example, primary care clinicians, pathologists, managers and policy-makers. The involvement of stakeholders at multiple stages is made clear in figure 2. This group will provide the content expertise essential for initial programme theory development and beyond. We will consult this group when focusing our review question and in assessing and developing candidate programme theories, to check that stakeholders agree that the theories under consideration are relevant and resonate with their experience.[86] Stakeholders may also suggest useful sources of evidence, and members of the group will be asked to provide feedback on iterations of refined programme theory as these are developed. Finally, the stakeholder group will be crucial in helping us to identify the most effective means of disseminating the results and recommendations that follow from the review.

### Patient and public involvement

Patients and the public will be involved throughout this review project via their inclusion as part of our stakeholder group. This means that patients will have the opportunity to help us prioritise the focus of this review and to develop and 'test' our programme theories as they

develop. In particular, we anticipate that patient input (as well as input from clinicians) will help us to identify and understand the important contexts, and reasoning at work whenever there is a decision to order tests (or otherwise). This input will help inform our searching and development of theory and ensure that the final refined programme theory resonates with patient experience.

### Step 1: develop initial programme theory

The first stage of a realist review is the development of an 'initial programme theory' which makes the first attempt to explain the phenomenon under examination. The development of this theory will be informed by two main processes: an informal scoping search of the literature and input from the stakeholder group.

Iterative, informal searching will be used to locate existing theories that are used to explain how and why overtesting and undertesting occur. This initial search stage will rely on a combination of more structured searching[88 89] and more emergent techniques such as reference and citation tracking ('snowballing') and personal contacts.[90] An inclusive approach will be used to screen documents found at this stage, with no limitations placed on type of study or document. Documents will be selected wherever there is an attempt to theorise about the causes of variation in test ordering, especially in relation to the circumstances in which such variation is most prevalent, and the reasoning of actors involved (even where such ideas are not identified formally as' theory').

This process may uncover informal 'folk theories'[91] attempting to explain the causes of variations in practice, and theories that underpin actual and proposed interventions designed to reduce the problem,[73] as well as potentially useful 'substantive' theory,[92] that is, established theory from any discipline which can help to explain the phenomenon. The stakeholder group will also be consulted to ensure that their content expertise is used to supplement the results of this early searching. Candidate initial programme theories will be presented, and stakeholders asked to provide feedback and commentary on their plausibility and 'fit' with their experience. Through this process, initial theory(ies) are likely to be refined and prioritised for the next stage of the review.

Work on this stage has begun and is ongoing. Initial search strategies focused on identifying relevant substantive theories are available in the online supplementary file. Figure 3 below illustrates the basis of an early set of initial programme theories, considering the 'decision to order test(s)' step from figure 1 above.

Initial exploration of the literature has uncovered a range of potentially useful substantive theory that could help to explain the mechanisms underlying the decision-making involved in test-ordering behaviour, including economic theory explaining oversupply and overconsumption in 'experts markets',[93] theories of decision-making that assume bounded rationality,[94] including regret theory[95] and threshold models[96] and several others.[97–99] These theories can be explored in relation to their ability to provide a useful lens through which to view this decision-making process in a realist fashion and explain observed outcomes. For example, 'regret theory' suggests the possibility of an underlying mechanism related to the estimation and minimisation of 'expected regret' in deciding to order a test or otherwise.

Another potentially valuable sources in the development of initial programme theory are those theories underlying interventions designed to reduce overtesting. Instead of assuming a complex decision-making process is happening, many such interventions seem based on the theory that test ordering is at least to some extent a habitual, normalised behaviour[100] and so seek to disrupt these habits. For example, interventions designed to increase barriers to test ordering[101 102] may create space

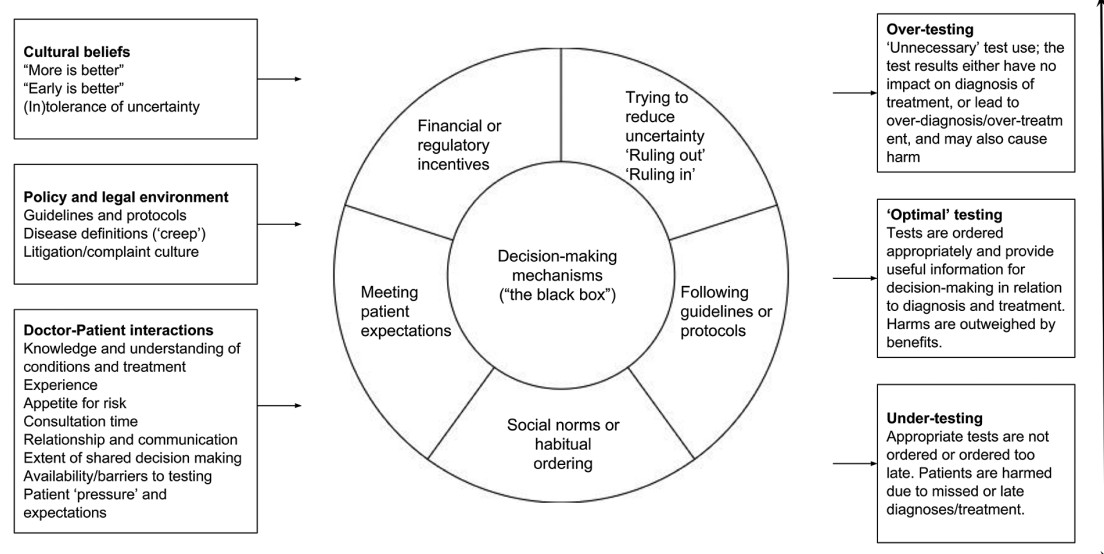

**Figure 3** Contexts, reasons for test ordering and range of outcomes.

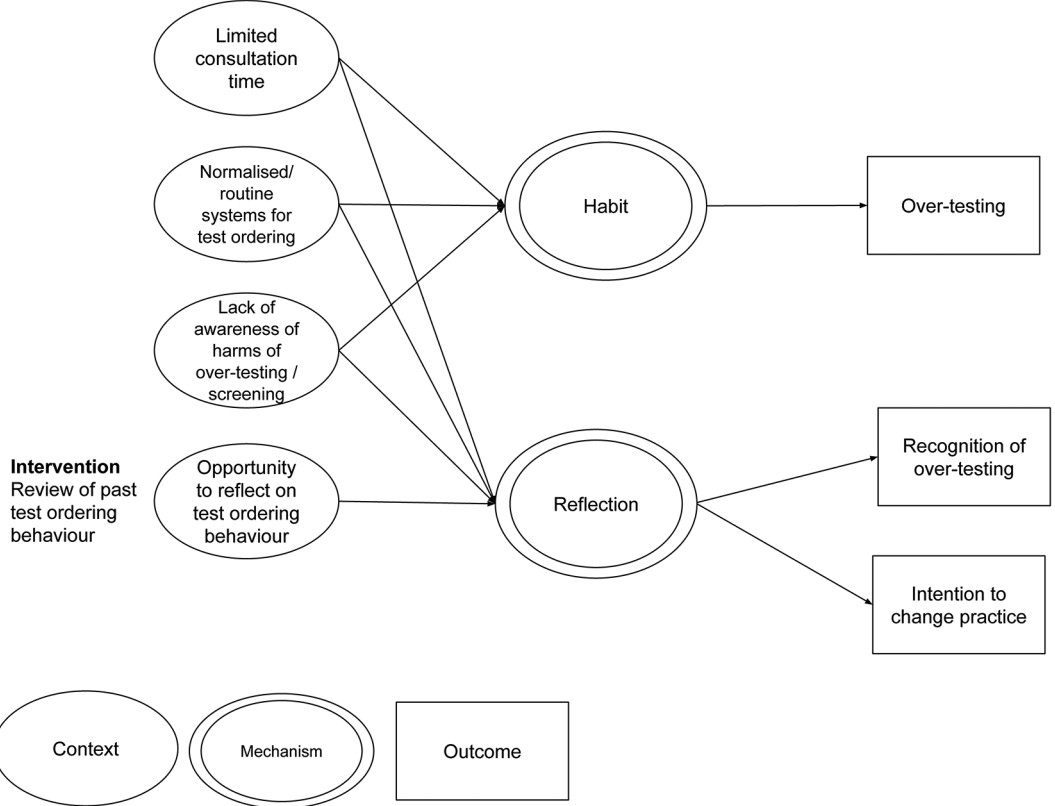

**Figure 4** Example CMOCs showing the possible effect of introducing reviews of test-ordering behaviour. CMOCs, context–mechanism–outcome configurations.

for doctors to consider whether a test is really necessary. Similarly, interventions designed to promote reflective practice[103–105] provide opportunities for doctors to reflect on their past test-ordering behaviour and outcomes and potentially change their behaviour in the future. Interventions based around computer-aided decision-support systems[106 107] may seek to replace old habits with new, evidence-based ones.

These initial theories can be conceptualised in a 'realist' fashion (ie, in the form of a CMOC), as illustrated in the hypothesised example in figure 4 above.

The candidate theories uncovered during searching will be considered by the project team alongside figure 3 to refine these initial CMOCs. These will be discussed with the stakeholder group and refined as necessary in light of these discussions and further reading. It is likely that a small number of candidate theories will be prioritised as a focus for the review, based on their greater importance and/or resonance with stakeholders.

### Step 2: searching for evidence

Secondary evidence gathered in cycles over the course of a realist review is iteratively interpreted and used to 'confirm, refute or refine' each aspect of a programme theory.[108] This evidence is sought from a wide range of sources and disciplines: there is no 'hierarchy of evidence' in a realist approach and so evidence may include quantitative and qualitative data, peer-reviewed articles, opinion and commentary, and grey literature like policy documents.[92]

The main systematic literature search(es) will be conducted with the aim of identifying relevant documents potentially containing data that can be used to develop or refine, refute or confirm, the initial programme theory(ies) chosen for testing.

A search strategy(ies) will be designed, piloted and executed by an information specialist (CD). A wide range of bibliographic databases covering multiple disciplines will be considered for searching, including MEDLINE, Embase, CINAHL, PsycINFO, PsycEXTRA, the Web of Science Core Collection, Scopus, ASSIA, IBSS, EconLit and Google Scholar. Sources of grey literature will be searched, including via web search engines. Free text and subject heading search terms will be chosen as appropriate, and the search strategy will be refined iteratively to achieve a balance of sensitivity and specificity. As for the informal search stage, 'snowballing' and other supplementary search techniques will be used to identify additional documents.[90]

Search results will be screened initially by title and abstract, with full text considered as a second step. A broad set of inclusion and exclusion criteria will be used to screen the results of the main search. These criteria will be finalised when the initial programme theory is confirmed, but are likely to include some or all of the following:

## Inclusion criteria

► All types of document.

► Any study design.

► Studies or documents that identify variation in test use, actual or potential underuse or overuse of tests, or are focused on areas of primary healthcare where undertesting or overtesting is a recognised problem.

► Studies or documents focused on primary care settings.

► If a particular type of test or specific test is chosen as a focus in consultation with the stakeholder group, searching may initially be limited to consider this area.

## Exclusion criteria

► Studies or documents focused on secondary care settings (though searches may be broadened later to consider additional settings if there is a dearth of literature focused on primary care, or where the stakeholder group or initial searches suggest common mechanisms may be in operation).

► Studies focused on imaging, genetic testing, fetal monitoring, near-patient testing, self-testing or home-based testing by patients (though searches may be broadened later, as above).

► Studies or documents focused on low-income and middle-income settings, where limited resources are likely to create very different contextual factors that are out of the scope of this review.

Screening of titles and abstracts will be undertaken primarily by the first reviewer (CD). An initial pilot batch of documents will be screened in duplicate by GW and the review team will meet to discuss discrepancies and assess agreement between the reviewers. Thereafter, a 10% random sample of search results will be screened by the second reviewer (GW) to check for consistency. Disagreements will be recorded and resolved via discussion in the project team.

As figure 2 illustrates, additional searching may be undertaken as required at later stages of the review, wherever the main search did not generate sufficient data to test programme theory (eg, if data on particular contexts or mechanisms were sparse), or in response to potential programme theory refinements. All such additional searches will be developed with an information specialist and screened as described above.

All searching and screening processes will be reported in full, including Preferred Reporting Items for Systematic Reviews and Meta-Analyses style flow diagrams,[109] to ensure transparency of evidence sources.

### Step 3: selection and appraisal

Following screening, documents will be selected on the basis of an assessment of their relevance (ie, whether some part(s) of the document can contribute to the refinement of programme theory) and rigour (ie, the trustworthiness of that data).[85] One reviewer (CD) will read all of the documents that met the inclusion criteria during screening and assess their ability to speak to some

aspect of the programme theory under consideration (ie, relevance). Relevant data from these documents will then be assessed for rigour.

The assessment of rigour in a realist review is not conducted at article or document level as in a 'traditional' review, since doing so may exclude documents containing relevant data[92] and even where a study as a whole is methodologically weak in terms of its own objectives, it may still contain 'nuggets' of useful data.[110] Instead, each piece of relevant contributing data will be judged according to its purpose in testing programme theory[85] and the methodology by which the particular piece of data was produced. This may involve the use of formal critical appraisal checklists suitable for different study types, but only as one part of determining trustworthiness. Different types of data will be subject to different judgements of methodological coherence and plausibility,[92] and the details of each assessment will be recorded in full to ensure that this process is transparent.

As with screening, a 10% random subsample of documents will be assessed by a second reviewer (GW) using the same criteria, with disagreements recorded and resolved via discussion in the project team. In anticipation of uncertainty in the case of some documents, the project team may also be called on to make assessments as a group.[79 111]

### Step 4: extracting and organising data

One reviewer will extract the main characteristics of each included document into an Excel spreadsheet. The full text of all of the documents will then be uploaded into the NVivo QRS International qualitative data analysis tool. One reviewer will then organise and classify this data, by annotating (coding) relevant data from each document according to its contribution to the developing programme theory.[85]

The initial phase of organising and coding data will be informed by any contexts, mechanisms and outcomes (or concepts not yet clarified as C, M or O) identified in the development of the initial programme theory. As data extraction progresses, organisation and coding is likely to evolve and include new concepts that reflect refinements to programme theory. As such, each document may be subject to several readings. As noted above, an individual document may include sections that contribute to several elements of programme theory. The use of data to refine programme theory will be recorded, to enable transparent reporting and the inclusion of relevant document extracts within the synthesis.[85] A 10% random subsample of documents that have been through the data extraction and organising process will be reviewed by a second reviewer (GW) to check for consistency, with disagreements recorded and resolved via discussion in the project team.

### Step 5: analysis and synthesis

In a realist review, analysis and synthesis of the selected data proceed in parallel, and will begin at the same point as document selection and appraisal for relevance and rigour, and data extraction and organisation.[75]

All three stages may thus proceed simultaneously (see figure 2), as data are chosen, assessed, annotated and organised according to its potential role in refining the developing programme theory.

This process will be iterative[75]: the programme theory will be refined in stages as more and more data are considered. The stakeholder group will be consulted at various points to obtain feedback on the focus and development of the programme theory and the project timeline will permit pauses in analysis and synthesis for this purpose, and to allow further searching to be undertaken where gaps in the available secondary evidence are found.

Pawson suggests that realist analysis and synthesis should be a process of 'juxtaposing, adjudicating, reconciling, consolidating and situating the evidence' in an effort to refine programme theory.[85] As such, data relating to different aspects of the programme theory will be collected together and considered alongside each other, such that an assessment of the strength of evidence supporting the arguments that underpin each aspect of that theory can be made. A process of retroductive reasoning will then be applied, so that refinements to programme theory are made on the basis of what can plausibly be inferred by all the data available. Retroductive reasoning will be used to build explanatory realist theory(ies). This involves an interpretive process of considering which underlying causal mechanisms must be at work to deliver the observed patterns out of outcomes. The approach involves moving back and forth between concrete observations and theory building, and hence between inductive and deductive reasoning.[112]

## Limitations and risks

An important potential limitation of this study will be the availability and contextual richness of the secondary evidence that is available.[75] Although initial scoping searches suggest that a significant amount of material on the subject of laboratory test ordering does exist, it is possible that this material will not describe contextual factors in great detail or include enough relevant information on which to build theory. We will attempt to address this problem by ensuring that comprehensive and wide-ranging searching is undertaken by an information professional, that supporting and related information for all included studies is located wherever it exists,[89] and by contacting authors to ask for further detail as required.

In addition, there are important limitations that are inherent to the nature of the realist review. In particular, there is a limit to how much ground a single review can cover and so this review will necessarily prioritise certain elements of the process within which test ordering takes place[75] and will inevitably have to set aside some potentially important factors for future research. The final output of the review will be a (refined) theory that attempts to illuminate important contextual factors and

underlying mechanisms; it is important to acknowledge that such theory can only ever represent partial knowledge that will be open to further refinement or refutation in the future.

## Outputs and dissemination

A variety of project outputs are planned, to meet the needs of different groups, including national and local policy-makers, leaders, employers and practitioners in primary care and pathology settings, and patients. To some extent, outputs will be guided by the review's conclusions and resulting recommendations that may have relevance in different contexts and at different levels.

The RAMESES reporting standards will be used to produce a complete and transparent report of this review—both for the funder and as a standalone publication.[87] The standalone publication will be for academic audiences and will be submitted as an article to a peer-reviewed journal. Other academic outputs will be prepared for presentation at relevant conferences (eg, 'Preventing Overdiagnosis,[113] International Realist Conference.[114]

The final refined programme theory and resulting recommendations will be presented to the stakeholder group (to include policy-makers, practitioners and patients) and their opinions will be sought to direct the dissemination strategy for these groups, with the aim of ensuring that important recommendations reach the appropriate decision-makers. We will endeavour in particular to reach policy-makers and researchers engaged in the development and evaluation of interventions designed to reduce variation in test ordering, in order that future work in this area can be informed by the new knowledge generated in this review. We envision the production of user-friendly and accessible summaries of the findings and our recommendations and the use of existing networks and social media to promote these outputs to help ensure maximum visibility.

**Contributors** CD conceived the study with input from GW. CD wrote the first draft of this manuscript and GW provided criticism and refinement. CD and GW approved the final version.

**Funding** CD is funded by a National Institute of Health Research (NIHR) Research Methods Programme Systematic Review Fellowship (NIHR-RM-SR-2017-08-018). GW and Trish Greenhalgh (TG) are supporting CD in this project. TG is partly funded by the National Institute for Health Research Biomedical Research Centre, Oxford, grant BRC-1215-20008 to the Oxford University Hospitals NHS Foundation Trust and the University of Oxford. GW is partly supported by the Evidence Synthesis Working Group of the NIHR School of Primary Care (Project Number 390).

**Disclaimer** The views expressed in this publication are those of the author(s) and not necessarily those of the NHS, the National Institute for Health Research or the Department of Health.

**Competing interests** CD and GW are both members of the Royal College of General Practitioners (UK) Overdiagnosis and Overtreatment Group. GW is an NHS general practitioner and joint deputy chair of the NIHR Health Technology Assessment Out-of-Hospital Panel.

**Patient consent** Not required.

**Provenance and peer review** Not commissioned; externally peer reviewed.

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
