## [Reviewer comments · BMJ Open]

ARTICLE DETAILS

TITLE (PROVISIONAL)	Explaining variations in test ordering in primary care: protocol for a realist review.
AUTHORS	Duddy, Claire; Wong, Geoffrey

VERSION 1 – REVIEW

REVIEWER	Michael Laposata University of Texas Medical Branch, Galveston, TX, USA
REVIEW RETURNED	04-May-2018

GENERAL COMMENTS	Major Criticisms 1. The realist review is highly complex and unfamiliar to most readers of this journal. It will be extremely valuable to present examples throughout the article that illustrate how the realist review improves understanding of how and why variation in laboratory testing ordering comes about.2. It is not clear what the stakeholders who will be recruited provide for the project. Specifically, what content expertise is needed for initial programme theory development that can be provided by the experts?3. Figure 2 is too complex to understand for anyone unfamiliar with the initial program theory. The second box from the top could have a footnote which provides an example or additional information to understand better the content of that box.4. Figures 3 and 4 are more suitable for a review article. It is not clear how the information it is report represents a protocol rather than a description of how the realist review program theory works.5. The information in figure 5 is background information and could be included in the introduction without a figure.6. The initial programme theory is so complex that the glossary of realist terminology does not help the reader understand the content of the article. The topic remains esoteric even with the glossary for all but the smallest percentage of readers. Minor Criticisms 1. There are too many references included for an article of this size.2. Figure 1 describes the process of ordering tests and interpreting the results. The legend improperly states that the figure shows the steps to over/under diagnosis. In summary, I believe that this article explaining variations in test ordering by primary care physicians as a protocol for a realist review is simply too esoteric for this journal, at least as written. Examples are needed to bring the theory into practice for those who are interested
--

	in trying to understand. From the understanding I acquired reviewing the article, this approach appears to leave out dozens of variables that should be included and were not to explain the variations in test ordering. With that in mind, the conclusions could be misleading, and therefore not helpful for both patients and for healthcare providers ordering laboratory tests.
--	---

REVIEWER	Harold S. Luft Palo Alto Medical Foundation Research Institute, United States
REVIEW RETURNED	18-May-2018

GENERAL COMMENTS	This protocol for a realist review addresses a very important topic likely to be highly informative to a variety of audiences. The general approach seems quite reasonable, but I outline below some concerns that the authors should consider as they move forward with this project. The focus of the project is not clear. Without a sharper (and probably narrower) focus, the project may be overwhelming and end only partially completed, or with an unevenness that calls into question the overall findings and interpretations. The beginning of the background section touches upon at least 5 topics: 1) variation in test ordering, 2) undertesting and overtesting, 3) [the impact of] campaigning, 4) overdiagnosis, and 5) the decision to order tests. While all are related, the "literatures" in which each topic is likely to be addressed, as well as the methods and data used (and implicitly the keywords associated with them) may vary substantially. When search terms rather than an expert-driven snowball approach is primarily being used to identify studies this may lead to missing important realms of research. 1. The notion of "variations" implies having numerator and denominator data. To study "variations in test ordering," most researchers would use moderate to large data bases to examine the rates of tests ordered for reasonably comparable patients. Some studies might stop with the evidence that there is variation, others may explore "factors" accounting for the variability, such as the years since the physician finished training, the physician's specialty, sex, or size of practice, or in the U.S., how they are compensated. 2. In contrast, studies focusing on "the decision to order" are likely to draw on a much wider range of data and approaches. Some might be ethnographic, observing in a few dozen encounters how a patient and physician discuss the advantages and disadvantages of ordering a test. Others might draw on physician or patient responses to selected hypothetical scenarios, perhaps purposefully varying the degree of pre-test uncertainty, or the consequences of not having the additional information. Yet others could use a large data base drawn from electronic health records in a large organization to examine physician and patient factors associated with the likelihood of ordering of a test in a specific encounter, e.g., a patient appearing with a given problem who is not well-known the physician and the physician is already running late when the appointment begins. Few of these studies, however, would offer much detail on organizational, payment, or other "contexts." 3. Studies of "over-diagnosis" are likely related to, but may delve more deeply into, what is meant by "over-diagnosis" and whether this is even something that can be measured objectively. Suppose a patient has a (possibly) unfounded concern about having a disease and this is resulting in anxiety. Is it over-testing to order a test, and is it overdiagnosis to get back a result that indicates little needs to be done, e.g., a low Gleason score prostate cancer? The test won't lead to treatments that would change his life expectancy, but it may address his anxiety. Alternatively, consider patients who refuse highly valuable screening due to fear of what they may learn, or religious beliefs about intervening to prevent something "natural" and thus
--

	appear to be undertested. Such situations could reflect underlying patient-specific issues that may be idiosyncratic, (a family friend who died of aggressive prostate cancer) and thus could be expected to average out across all the patients seen by a primary care physician (PCP). Exploring these issues would be irrelevant except that they may explain why all PCPs fail to be completely adherent to the "guidelines." However, such patient "preferences" may also be associated with certain ethnic or religious groups. If so, it is plausible that some PCPs will have a disproportionate share of such patients . They would then appear to be consistent "over (or under) diagnosers." 4. Studies addressing under and overtesting, as well as the effects of "campaigns" to address those issues, implicitly focus on specific tests for which there is some consensus (hopefully evidence-based) regarding when a test should, and should not, be done. At least in the U.S. this is not always a "settled issue" e.g., with regard to PSA tests and the frequency of mammograms, where various professional societies disagree with the guidelines. I doubt many studies relevant to categories 2-4 would have "variations" in their text, let alone the titles of the papers. Unless done very carefully, searches may differentially access the underlying literature relevant to each of these questions. Each of these broad areas is well worth extensive investigation, but I think the underlying conceptualization of the issues, as well as the ways in which the review should be undertaken will differ based on the topic. Doing all simultaneously will be a major project. Unless there are substantial resources and time available, I suggest narrowing the scope. If there is lack of clarity at the outset, it may be much more difficult to achieve a clear and convincing review. The current protocol envisions engaging stakeholders early. This may solve the problem, but I encourage an explicit process for deciding on whether to narrow the focus, and which topic(s) should be addressed. Minor points: Pg 8 Para 2 You may also want to consider, especially in the context of "campaigns," the role of various "nudges" to shape behaviors. (See Thaler and Sunstein, Nudge: Improving Decisions About Health, Wealth, and Happiness, 2009. Pg 9 lines 32-34 The criteria here seem focused on just a subset of the questions introduced at the beginning. The inclusion criteria and the problem to be addressed need alignment. Pg 11 line 14 It is unclear when in the process the 10% "second readings" will occur. Might it be better to do this as the process is unfolding, e.g., every 10th document would be read by GW? (This can be rough, i.e., not "truly random"—precise sampling isn't the issue here.) Differences would in the assessment by CD and GW then be discussed, thereby implicitly, or perhaps explicitly, changing the ways that CD is coding the next batch of documents. Keeping track of those discussions, moreover, can be important because there may be issues of sufficient import to go back and "recode" some of the earlier documents.
--	--

VERSION 1 – AUTHOR RESPONSE

Reviewer 1

We thank Michael Laposata for his time and thoughtful comments on the protocol. We have made a number of changes to the manuscript and provide explanations below in response to each of his

criticisms. We believe that this topic is highly relevant for BMJ Open readers and that our detailed realist review protocol (which are already regularly published in this journal) is something that readers of this journal will find interesting. We fully acknowledge that the review cannot cover everything that might affect test ordering, but instead must take a particular 'cut' through the available data.[1,2] The chosen 'cut' will be informed by both the availability of data, and the input of the patients, doctors, managers and policy makers in the stakeholder group (either via their prioritisation of particular areas, or identification of areas of focus most amenable to change and improvement). This approach is inherent in the nature of realist review, following established quality standards,[3] and it is on this basis that we will aim to generate useful theory that can inform the development of future interventions and policies.

In response to his major criticisms:

1. Realist review is growing in popularity and becoming much more established as an approach in health research. BMJ Open has published over 50 realist protocols, reviews and evaluations to date (GW is an author on many of these). We hope that BMJ Open readers are growing in familiarity with the realist approach, and that our protocol will help to add to this understanding. We have made some adjustments to the background information provided on the realist approach to increase its accessibility for any readers who may not have come across it before (in section 'Realist review').

2. The stakeholder group will provide a number of valuable functions during the review process, described on page 7 (lines 14-23 in the original manuscript). The timing of stakeholder involvement is indicated in Figure 2, which outlines the processes we will follow in conducting the review. The RAMESES publication standards for realist syntheses emphasise the need to 'draw on external stakeholder expertise' in focusing the review question and maximise the relevance of the review for end-users.[3] We have expanded the information provided on page 7 to highlight the specific roles that stakeholders will play at each stage in the review. This includes in the provision of 'content expertise', meaning we will consult the group in order to assess whether candidate programme theories resonate with their own experience as patients, doctors, managers and policy makers in relation to variations in test ordering (see section "Stakeholder involvement").

3. Figure 2 provides a flow diagram-style illustration of the processes undertaken in a realist review, and inclusion of a similar diagram is common in other protocols.[see, e.g. ,4] The figure's purpose is described on page 6 (lines 54-55 in the original manuscript). The second box from the top highlights the two major elements of the work that will be undertaken to develop the initial programme theory, i.e. informal literature searching and consultation with stakeholders. This step is described in much more detail on page 7 (lines 25 onwards in the original manuscript). To make clear that this explanation matches this part of the diagram, we have made a small change to Figure 2, adding the word 'step' in each box referring to a step of the review process to match how these stages are described in the text.

4. Figures 3 and 4 are provided here for two purposes. First of all, to illustrate to the reader what 'initial programme theory' development might look like; secondly, to make transparent the thinking processes underway at this early stage of the review. Transparency is a crucial feature for realist review,[3] ensuring clear visibility of decision making processes and theory development, and so here we are laying these processes bare. It is fully anticipated that the ideas presented in these figures will evolve as the review progresses, in light of stakeholder feedback and available data (noted at the end of the section, 'Step 1...').

5. Figure 5 is included only as a useful visual representation and we are content for the editor to leave it out if preferred.

6. We hope that the glossary acts as a useful reference guide for readers less familiar with realist terminology, who might be interested to explore the definitions of terms. However we have also made every effort to ensure that the text is readable without continual reference to the glossary, by explaining realist terms in context as they are used. We hope that the adjustments made to the background section on realist review (mentioned in point 1 above) have helped here.

Minor criticisms

1. The article is well-referenced due to the large volume of existing literature already identified in the development of the protocol. The referencing is an accurate reflection of the work that has been completed during development of the protocol, and as such, contributes to the overall transparency of the review process.

2. Thank you for pointing this one out - this diagram was replaced and the title not updated. We have changed this now to, "Figure 1: Steps taken in test ordering decisions".

Reviewer 2

We would like to thank Harold S Luft for his time and contribution to the protocol review. As with the comments provided by the first reviewer, we have taken his comments on board and outline our changes and responses below. In relation to his major concern about the focus and potential breadth of the project, we can offer the reassurance that significant time and resource is available for the review (c. 1.5 - 2 years between September 2018 and August 2020), and that we anticipate that a sharper focus will evolve as the review progresses. An iterative approach to focusing is inherent in the realist approach[2,5,6] and our focus will be directed by both the availability of data, and the input of the patients, doctors, managers and policy makers in the stakeholder group (either via their prioritisation of particular areas, or identification of areas of focus most amenable to change and improvement).[3]

The types of studies anticipated by this reviewer in points 1-4 (i.e. studies based on large data sets, ethnographic or simulated studies in relation to decision making, wider discussions of definitions in relation to overdiagnosis phenomena and studies reporting the effects of campaigns and other interventions designed to reduce variation in practice) are exactly the types of study that we anticipate finding and utilising in the course of this review. Preliminary searching has located some of the documents as listed above, and we expect further diverse literature (including grey literature) may be included, too. We acknowledge that this literature will include many different conceptualisations of the problem of variation in test ordering and aim to address this with our first review question, "How are 'undertesting' and 'overtesting' conceptualised in the literature?" (page 6, line 43 of the original manuscript).

We agree that the search strategies used to locate such a diverse range of material will need to be carefully planned. CD is an information specialist, with significant experience of developing complex search strategies. Our plan for searching is outlined in the protocol (beginning on page 8, line 51 onwards in the original manuscript) and notes that the strategy will employ a range of techniques and sources for exactly the reasons identified by the reviewer. It is worth noting that the aim in a realist review is not exhaustiveness (which would be impossible for a project of this size), but rather to reach 'theoretical saturation', i.e. to identify sufficient material to inform and refine the development of theory. We are interested in a wide range of study types and other material and will seek to identify relevant contextual information wherever we can, allowing us to theorise about the 'mechanisms' in play. We have acknowledged that the availability and richness of data may be an important limitation for our study (page 3, line 12 of the original manuscript).

In response to the minor points raised:

1. We agree, and certainly anticipate that 'nudge' style interventions (and theory) will be discussed in the literature and may well make their way into this review.

2. The inclusion and exclusion criteria set out on page 9 aim to ensure that we retrieve and include a diverse range of literature on the subject of test ordering in primary care (to include many study types as outlined above). The criteria do not cover some of the issues outlined in the background section (e.g. campaigns, overdiagnosis more generally) - these issues are covered in this background to provide an overall context for the project, but are without its scope (unless as they relate specifically to test ordering).

3. We have made an adjustment to the proposed duplicate checking processes in the review (described on page 10, lines 3-6 and 42-44, and page 11, lines 12-15 of the original manuscript) based on this very useful feedback. Although time constraints do not permit GW to screen every 10th document, we propose that an initial 'batch' of documents will be screened/assessed/extracted in duplicate at the beginning of these processes, effectively introducing short 'piloting' periods including discussion between reviewers, to ensure agreement in understanding and coding decisions etc. All such discussions will be recorded in the interests of transparency.

Formatting amendments

1. The supplementary file has been uploaded in PDF format.

2. Patient and Public Involvement - a section has been added to the methods section (after the section on 'Stakeholder involvement').

References

1 Wong G. The Internet in Medical Education: A Worked Example of a Realist Review. In: K.Hannes CL, ed. *Synthesizing Qualitative Research: Choosing the Right Approach*. Chichester: : John Wiley & Sons, Ltd. 2011. doi:10.1002/9781119959847.ch5

2 Pawson R, Greenhalgh T, Harvey G, et al. *Realist synthesis: an introduction*. 2004. https://www.researchgate.net/profile/Gill_Harvey/publication/228855827_Realist_Synthesis_An_Introduction/links/0fcfd507f0b7cbb2ce000000/Realist-Synthesis-An-Introduction.pdf (accessed 7 Dec 2017).

3 RAMESES Project Team. *RAMESES Quality Standards For Realist Synthesis (for researchers and peer-reviewers)*. 2014. http://www.ramesesproject.org/media/RS_qual_standards_researchers.pdf (accessed 7 Dec 2017).

4 Wong G, Brennan N, Mattick K, et al. Interventions to improve antimicrobial prescribing of doctors in training: the IMPACT (IMProving Antimicrobial presCribing of doctors in Training) realist review. *BMJ Open* 2015;5:e009059. doi:10.1136/bmjopen-2015-009059

5 Pawson R, Greenhalgh T, Harvey G. Realist review - A new method of systematic review designed for complex policy interventions. *J Heal Serv Res Policy* 2005;10:21–34. <http://ovidsp.ovid.com/ovidweb.cgi?T=JS&PAGE=reference&D=emed10&NEWS=N&AN=40993882>

6 Pawson R. *Realist Synthesis: New Protocols for Systematic Review*. In: *Evidence-based policy: a realist perspective*. London: : SAGE Publications Ltd 2006. 73–104. doi:10.4135/9781849209120

VERSION 2 – REVIEW

REVIEWER	Harold S. Luft Palo Alto Medical Foundation Research Institute, United States
REVIEW RETURNED	05-Jul-2018
GENERAL COMMENTS	Revisions all make sense. No additional comments.